# Risk of Post-COVID-19 Uveitis and Risk Modification by Vaccination: A Nationwide Retrospective Cohort Study

**DOI:** 10.3390/vaccines12060631

**Published:** 2024-06-06

**Authors:** Jiyeong Kim, Seong Joon Ahn

**Affiliations:** 1Department of Pre-Medicine, College of Medicine, and Biostatistics Laboratory, Medical Research Collaborating Center (MRCC), Hanyang University, Seoul 04763, Republic of Korea; kimzi@hanyang.ac.kr; 2Department of Ophthalmology, Hanyang University Hospital, Hanyang University College of Medicine, Seoul 04763, Republic of Korea

**Keywords:** COVID-19, vaccination, infection, uveitis, risk, nationwide cohort

## Abstract

This study aimed to evaluate the risk of uveitis, one of the most common ocular manifestations of COVID-19, in individuals with a history of uveitis and COVID-19 infection while discriminating the effects of COVID-19 infection and vaccinations. We analyzed nationwide data from 235,228 individuals with a history of uveitis prior to COVID-19 infection and evaluated incidences and hazard ratios (HRs) of post-COVID-19 uveitis for different post-infection periods, including early- (within 30 days) and delayed-onset ones. The cumulative incidences of post-infection uveitis at 3, 6, and 12 months were calculated as 8.5%, 11.8%, and 14.0%, respectively. The HR of post-COVID-19 uveitis was 1.21 (95% confidence interval [CI]: 1.07–1.37) and was particularly higher in the early-onset period (1.42, 95% CI: 1.24–1.61). Vaccinated individuals showed a modestly elevated risk of uveitis relative to pre-infection, while unvaccinated ones exhibited substantially higher risks in the early-onset period: the HR of post-infection uveitis before vaccination was 3.61 (95% CI: 1.35-9.66), whereas after vaccination, it was 1.21 (95% CI: 1.05–1.39). COVID-19 infection was associated with a higher risk of uveitis, which was mitigated by vaccination. Vigilance in the monitoring of uveitis is warranted for recently COVID-19-infected individuals with a history of uveitis, particularly unvaccinated individuals.

## 1. Introduction

Uveitis, characterized by the inflammation of the uvea, is a potentially sight-threatening ocular condition that encompasses a spectrum of inflammatory disorders affecting various structures within the eye [1]. It is a significant cause of visual impairment worldwide, accounting for a considerable proportion of ocular morbidity and blindness [1,2]. With its multifactorial etiology and diverse clinical presentations, uveitis poses diagnostic and therapeutic challenges to clinicians, necessitating a comprehensive understanding of its risk and associated factors [1,2].

The COVID-19 pandemic, caused by the novel coronavirus SARS-CoV-2, has prompted extensive research into its systemic manifestations, including its potential impact on ocular health [3]. During the ongoing COVID-19 pandemic, emerging evidence suggested a potential association between COVID-19 infection and ocular manifestations [4,5,6]. Indeed, COVID-19 has been linked to various ocular complications, ranging from ocular surface inflammations, such as conjunctivitis, to more severe intraocular inflammation, such as uveitis [3,4,5,6]. Understanding the impact of COVID-19 infection on uveitis recurrence is crucial for elucidating the interplay between viral infections and ocular inflammation and thereby guiding clinical management strategies and preventive measures.

Previous studies have provided insights into the relationship between COVID-19 and uveitis. While some have reported uveitis to be one of the most common complications of COVID-19 infection [7,8], others have reported post-vaccination cases as well [9,10,11,12,13]. Among these studies, several population-based ones revealed an association between COVID-19 vaccination and uveitis [10,12,13]. However, discerning the distinct effects of COVID-19 infection and vaccination on uveitis recurrence remains a challenge, necessitating further research to disentangle these complex relationships, especially considering the substantial overlap between vaccinated individuals and those who subsequently become infected. Uveitis is particularly concerning for individuals with a history of uveitis, as several studies have shown uveitis history to be a significant risk factor for uveitis following COVID-19 vaccination [12,14].

Therefore, the primary objective of this study was to comprehensively analyze the risk of uveitis recurrence following COVID-19 infection in people with a history of uveitis while meticulously distinguishing between the effects of COVID-19 infection and vaccinations. Leveraging nationwide data and employing a rigorous methodology, this investigation aimed to obtain valuable insights into the association of uveitis with COVID-19 infection and vaccination, thereby informing clinical practice and guiding public health interventions for COVID-19 patients and the vaccinated population.

## 2. Materials and Methods

### 2.1. Study Population

This is a retrospective study on a nationwide cohort with a history of uveitis and COVID-19 infection observed for the study period between 1 January 2015 and 31 December 2022. We utilized the combined data from two databases: the Korean National Health Insurance Service (NHIS) database, which includes the entire Korean population diagnosed with uveitis between 1 January 2015 and 19 January 2020 (the day before the first COVID-19 infection in Korea), and the Korea Disease Control and Prevention Agency (KDCA) database, which provides details on COVID-19 infection and vaccinations, including types and timings of COVID-19 vaccine doses administered, as well as the diagnosis and timing of COVID-19 infections between 8 October 2020 to 31 December 2022 (the end of our study). Other information on demographic data, inpatient and outpatient healthcare visits, prescription medications, and diagnoses were provided by the NHIS. Diagnostic codes for uveitis identification were derived from the Korean Standard Classification of Diseases, 7th and 8th Revisions, in alignment with the International Statistical Classification of Diseases and Related Health Problems, 10th Revision (ICD-10), whose codes were established for the identification of uveitis in prior studies [14,15].

Among the individuals with a history of uveitis and COVID-19 infection, those for whom the date of their COVID-19 infection or vaccination was not recorded were excluded. Additionally, individuals infected with COVID-19 within 1 month of vaccination, either before or after, were excluded to eliminate the combined acute effects of recent vaccination and infection exerted on the individuals, for whom discriminating one from the other would have proved impossible. After the exclusion, the study population consisted of the entire Korean population with a prior history of both uveitis and COVID-19 infection, for whom the dates of infection and vaccination, separated by at least a 1-month interval, were both recorded.

For comprehensive analyses on the effect of COVID-19 infection, its vaccination, and a combination of both, the individuals were categorized as follows. First, those with COVID-19 infection were separated into those with both COVID-19 vaccination and infection and those with infection only. Second, the individuals with both vaccination and infection were divided into two groups: infection before 1st vaccination and infection following 1st vaccination. Figure 1 provides a graphical summary of the inclusion/exclusion criteria and our subject categorization scheme.

### 2.2. Definitions and Outcomes

Uveitis was categorized based on the timing of onset following COVID-19 infection and classified according to anatomical type and etiology. Early-onset and delayed-onset post-infection uveitis were defined as occurring within 30 days of the most recent infection and thereafter, respectively. The observation period was also stratified according to COVID-19 vaccination and infection. The pre-vaccination or pre-infection period was defined as the period preceding any COVID-19 vaccination or infection event, whichever came first. The post-infection-only period was characterized according to the time following the first COVID-19 infection but preceding the first COVID-19 vaccination. Conversely, the post-vaccination-only period indicated the time following the first vaccination but preceding the first COVID-19 infection event. Lastly, the post-vaccination and infection period encompassed that following both vaccination and infection, irrespective of the sequence of occurrence. Uveitis types were classified as anterior or non-anterior and as infectious or non-infectious using the relevant KCD-8/ICD-10 codes, consistent with previous studies on the same database (NHIS). The definitions and specific codes used for this study are outlined in Appendix A [13,14,15].

Demographic characteristics, including age and sex, as well as comorbidities, such as hypertension, diabetes, and rheumatic diseases, were extracted from the database. Information on the timing of COVID-19 infection relative to vaccination status and details on vaccinations, including the number of vaccination doses received and the type of vaccine administered, were also collected.

### 2.3. Analyses

Descriptive statistics were used to summarize the demographic and clinical information. Continuous variables are presented herein as means (SD) or medians (IQR), whereas categorical variables are expressed as frequencies and percentages. Cumulative incidence of post-infection uveitis was determined using Kaplan–Meier curves, and the incidences were assessed for different post-vaccination periods (3 months, 6 months, and 1 year). Rates of pre- and post-vaccination uveitis were computed by person-months and hazard ratios (HRs), and corresponding 95% confidence intervals (CI) were calculated to evaluate the risk of post-infection uveitis, using the pre-infection period as the reference. For this risk assessment, Cox proportional hazards models were utilized while adjusting for age, sex, and comorbidities. In addition to the risk for the overall subjects, separate analyses were conducted for subgroups stratified by the status and timing of COVID-19 vaccination relative to the onset of COVID-19 infection. The significance of all p-values was determined using two-tailed tests, with statistical significance defined as *p*  <  0.05. Statistical analyses were conducted using SAS Enterprise Guide version 7.1 and R version 4.0.3.

This study followed the principles of the Declaration of Helsinki and was approved by the Institutional Review Board of Hanyang University Hospital (IRB File No. 2023-02-043). The requirement for informed consent was waived because of the retrospective design and anonymized data. This study conforms to the STROBE Reporting Guidelines for Cohort Studies.

## 3. Results

### 3.1. Clinical Characteristics of the Study Population

Among the 236,710 individuals with a history of uveitis and COVID-19 infection, 235,228 were included in our analyses after applying the exclusion criteria (Figure 1). The cohort comprised 220,745 individuals who had both a COVID-19 vaccination and infection and 14,483 who had the infection only. Table 1 provides an overview of the demographic characteristics and comorbidities among the study population consisting of individuals with a history of uveitis and COVID-19 infection. Both the crude number of individuals and their corresponding percentages are included. The mean (SD) age of the participants was 55.6 (19.6) years, highlighting a diverse age distribution. The majority of participants fell into the 40–79 age range, with the largest proportion being in the 60–79 age group (38.0%), followed by the 40–59 age group (29.0%). In terms of sex distribution, there were slightly more females (53.9%) than males (46.1%) in the study cohort. Comorbidities were prevalent among the participants, with rheumatic diseases being the most common (68.9%), followed by hypertension (38.5%) and diabetes (28.9%). Regarding the number of infection events, the vast majority experienced only one COVID-19 infection event (98.2%), while only a very small proportion experienced two (1.8%) or three events (0.0001%). The timing of infection in relation to vaccination status varied, with a notable percentage having incurred the infection after completing their second (or subsequent) vaccination (92.2%), whereas 7.0% of individuals experienced a flare before the first vaccination during the observation period.

Vaccination coverage was high among the study population, with 93.8% of individuals having received their first dose and 93.1% their second dose. Moreover, a considerable proportion of participants received a third dose (75.3%), and some even received a fourth or further dose (31.8%). The Pfizer-BioNTech (BNT162b2) vaccine was the most commonly administered for the first vaccination (52.0%), followed by the AstraZeneca (ChAdOx1) vaccine (35.4%), Moderna (mRNA-1273) vaccine (10.0%), and Janssen (Ad26.COV2.S) vaccine (2.5%).

### 3.2. Risk of Post-COVID-19 Uveitis in Overall Subjects and Subgroups According to COVID-19 Infection and Vaccination

Table 2 provides the HRs for pre- and post-COVID-19 uveitis in the overall subjects, along with the corresponding CI. In the pre-infection period, there were 175,325 events, resulting in a rate of 0.022 per person-month. Following COVID-19 infection, there were 1,839 events, yielding an HR of 1.21 (95% CI: 1.07-1.17), indicating a significantly increased risk of uveitis relative to the pre-infection period. Further analysis revealed that early-onset uveitis had an HR of 1.42 (95% CI: 1.24-1.61), while delayed-onset uveitis had an HR of 1.09 (95% CI: 1.03-1.14), demonstrating a varying degree of risk depending on the timing of uveitis onset relative to COVID-19 infection.

Table 3 presents the HRs of pre- and post-COVID-19 uveitis in individuals who had received COVID-19 vaccinations at the time of COVID infection as compared with those who had not. Among those who were already vaccinated before COVID-19 infection, the pre-infection period had 2882 events, with a rate of 0.023 per person-month. Following infection and until the next vaccination or re-infection, there were 214 events, with a similar rate of 0.023 per person-month, resulting in an HR of 1.21 (95% CI: 1.05–1.39). In contrast, among those not vaccinated before COVID-19 infection, the pre-infection period had 1345 events, with a rate of 0.026 per person-month. The post-infection period until the first vaccination showed a substantially elevated rate of 0.049 per person-month, resulting in a remarkably high HR of 2.07 (95% CI: 1.42–3.02), indicating a substantially increased risk of uveitis post-infection in unvaccinated individuals. An analysis by onset timing showed different HRs for early (3.61, 95% CI: 1.35–9.66)- and delayed (1.94, 95% CI: 1.29–2.90)-onset uveitis.

Figure 2 plots the HRs of pre- and post-vaccination or infection uveitis, thus providing insights into the risk of uveitis relative to vaccination status and COVID-19 infection in the overall subjects. In the pre-vaccination or infection period, there were 172,431 events, with a rate of 0.022 per person-month. During the post-vaccination-only period, there were 2882 events, with a slightly elevated rate of 0.023 per person-month, resulting in an HR of 1.18 (95% CI: 1.13–1.22), indicating a significantly increased risk of uveitis post-vaccination. Similarly, in the post-infection-only period, with a rate of 0.028 per person-month, the HR was 1.61 (95% CI: 1.37–1.89), suggesting the highest risk of uveitis. In the post-vaccination and infection period, with a rate of 0.021 per person-month, the HR was 1.10 (95% CI: 1.05–1.15), indicating that the HR was lower than that in the post-infection-only period but was similar to that in the post-vaccination-only period.

In the subgroup analyses, according to age and sex, a similar trend of the highest risk of uveitis in the post-infection-only period was noted across all age and sex groups. However, a slight increase in the risk of post-COVID-19 uveitis was observed in older age groups (Appendix A), whereas no remarkable difference in post-infection uveitis was observed between male and female patients (Appendix A).

### 3.3. Cumulative Incidences of Uveitis over Different Time Periods and Analysis of Anatomy and Etiology

Table 4 presents the cumulative incidences of post-COVID-19 uveitis over different time periods, shedding light on the evolving risk of uveitis following COVID-19 infection. Within 3 months post-infection, 19,946 cases of uveitis were reported, representing 8.5% of the study population. This proportion increased to 27,729 (11.8%) at 6 months and further to 32,795 (14.0%) at 1 year post-infection. When considering uveitis types, the anterior type accounted for the majority of cases across all time periods: 16,456 (7.0%) at 3 months, 22,943 (9.8%) at 6 months, and 27,275 (11.6%) at 1 year post-infection. In contrast, the non-anterior type exhibited lower cumulative incidences: 3490 (1.5%), 4,786 (2.0%), and 5520 (2.4%) cases at 3 months, 6 months, and 1 year, respectively.

Table 5 highlights the association between anatomic types and disease processes in pre- and post-COVID-19 uveitis cases. A significant association was observed between the anterior and non-anterior types of uveitis pre- and post-infection (*p* < 0.001). Specifically, 99.0% of the individuals (24,080 of 24,343) with anterior uveitis before COVID-19 infection had anterior uveitis during the post-infection period, whereas 27.4% of those (1837 of 6697) with pre-infection non-anterior uveitis had post-infection anterior uveitis, the difference of which was statistically significant (*p* < 0.001). Similarly, a significant association was found between infectious and non-infectious disease processes (*p* < 0.001). These findings underscore the impact of COVID-19 infection on the distribution of uveitis types and disease processes, indicating that most incidences of post-COVID 19 uveitis were of the anterior or non-infectious type.

## 4. Discussion

The findings of this study revealed clues to the relationship between COVID-19 infection, vaccination, and uveitis recurrence. The observed increase in the incidence of uveitis following COVID-19 infection highlights the potential ocular manifestations of the virus. The higher HRs for early-onset uveitis suggest a possible temporal association between COVID-19 infection and uveitis recurrence. The predominance of anterior and non-infectious uveitis following COVID-19 infection in our study provides clinically important information for managing post-COVID-19 uveitis effectively.

The clinical characteristics outlined in Table 1 offer valuable insights into the study population of individuals with a history of uveitis and COVID-19 infection. The high vaccination coverage observed among the study population reflects the widespread acceptance and uptake of COVID-19 vaccines, with the Pfizer-BioNTech vaccine being the most commonly administered. This also suggests that distinguishing between the impact of vaccination and COVID-19 infection on uveitis recurrence among the included patients with COVID-19 infection is crucial, considering that a vast majority of these patients received vaccination during the study period. Moreover, with 68.9% of the included patients presenting rheumatic diseases, it is imperative to recognize the potential interplay of underlying immune-mediated mechanisms with COVID-19 infection in the manifestation of post-infection uveitis.

The data on the risk of post-COVID-19 uveitis among the different subject groups and vaccination statuses are presented in Table 2 and Figure 2. They demonstrate a significant association between COVID-19 infection and increased risk of uveitis recurrence, as evidenced by the elevated HRs observed in individuals following infection relative to the pre-infection period. Particularly, early-onset post-infection uveitis exhibited greater HRs, suggesting that the period should be carefully monitored for uveitis recurrence in individuals with a history of uveitis. Additionally, the analysis stratified by vaccination status (Table 3) revealed intriguing patterns, with vaccinated individuals showing a modestly elevated risk of uveitis post-infection relative to the pre-infection period (HR 1.21). Conversely, unvaccinated individuals exhibited a higher risk of uveitis post-infection (HR 2.07), underscoring the importance of vaccination in mitigating the risk of uveitis recurrence following COVID-19 infection.

Furthermore, our findings for the different post-infection and vaccination periods elucidate the differential impact of vaccination and infection on the risk of post-COVID-19 uveitis. The significantly elevated HRs during the post-infection-only period (1.61) highlight the acute, severe intraocular inflammation, as is typically observed with systemic inflammation following COVID-19 infection. Conversely, the slightly elevated HRs observed during the post-vaccination-only period (1.18) suggest an association between vaccination and uveitis, albeit to a lesser extent compared with infection. This is consistent with previous findings showing significant associations between COVID-19 vaccination and uveitis recurrence for people with a history of uveitis [12,16]. The slightly elevated HR during the post-vaccination and infection period (1.10), similar to that during the post-vaccination-only period, may indicate a much-mitigated host response after COVID-19 infection in vaccinated individuals, leading to a reduced risk of uveitis recurrence.

While concerns have been raised regarding uveitis recurrence following vaccination, our results suggest that vaccination may actually mitigate the risk of uveitis recurrence in individuals with a history of uveitis who contract COVID-19. These findings underscore the importance of balanced risk assessment and informed decision-making regarding vaccination, particularly for more vulnerable populations, such as those with recurrent uveitis. Therefore, our study contributes to the ongoing dialogue surrounding vaccination strategies for uveitis patients and potentially suggests the role of vaccination in alleviating the overall burden of COVID-19-related complications, including post-infection uveitis.

The cumulative incidences of post-COVID-19 uveitis, as illustrated in Table 4, reveal a notable increase in the proportions of uveitis cases observed at 3 months, 6 months, and 1-year post-infection. This data highlights the prolonged but mainly short-term impact of COVID-19 infection on uveitis, as more than half of the recurrence cases had uveitis incidence during the initial 3 months. Moreover, the predominance of anterior and non-infectious uveitis in post-COVID-19 flares across all time periods (Table 4), significant compared with pre-COVID-19 flares (Table 5), underscores the distinct anatomical distribution and etiology of uveitis following COVID-19 infection. However, the presence of non-anterior and non-infectious uveitis underscores the need for the comprehensive monitoring for diverse uveitis phenotypes in clinical practice, despite the lower cumulative incidences. Also, further research is warranted to elucidate the underlying mechanisms driving the predominance of the anterior and non-infectious types, as well as to inform targeted interventions aimed at the successful treatment of post-COVID-19 uveitis.

Moreover, comparing the risk of uveitis recurrence associated with COVID-19 with that associated with other systemic viral infections could offer valuable insights into the distinct characteristics of post-COVID-19 uveitis. Although recurrent uveitis following other viral infections has been documented, existing reports predominantly focus on infectious uveitis resulting from viral infections, particularly those of the Herpesviridae family, such as cytomegalovirus, herpes simplex virus, and varicella-zoster virus [17,18,19,20]. One population-based study showed a significant association between herpes zoster and the subsequent risk of anterior uveitis, with a hazard ratio (HR) of 1.67 during the 1-year follow-up period following zoster infection, increasing to 13.06 for herpes zoster ophthalmicus [18]. However, systematic evaluations of post-infection uveitis in the context of other viral infections have been limited because of the smaller patient populations studied, unlike for COVID-19, which has affected a large number of individuals globally as a pandemic. Future studies should investigate how the risk of uveitis recurrence is associated with COVID-19, compared to that of other viral infections, including herpes viruses or other viruses known to induce ocular inflammation [17,18,19,20]. Understanding the relative risk posed by COVID-19 compared to other viral infections could illuminate potential shared mechanisms or distinct pathophysiological pathways underlying post-viral uveitis. Furthermore, comparative analyses may inform clinical management strategies and guide tailored preventive measures specific to different viral etiologies.

Several limitations of this study should be acknowledged, including its retrospective design and reliance on diagnostic codes in the NHIS database for data collection, which may lead to the underestimation of outcomes due to under-reporting. Misclassification bias may be another important limitation of population-based studies using health-claim databases [21]. The post-vaccination period might be heterogeneous in terms of uveitis risk, owing to subsequent vaccinations, despite having a smaller effect than the first one in previous studies [12,13,16]. However, the post-vaccination period in this study, rather than the post-first vaccination or post-second vaccination period, is simple and straightforward and represents the effect of the most significant event in terms of vaccine-associated uveitis, the first vaccination. Further, we could not conduct any assessment of de novo post-COVID-19-infection uveitis (by comparison with non-COVID-19-infected individuals), which also would be necessary for a complete understanding of the effect of COVID-19 infection on uveitis. Moreover, due to limited available data on the clinical characteristics and other details of uveitis and COVID-19 infection, we could not adjust for several confounders, such as the presence of, or specific details on, immunomodulatory therapy. Additionally, the generalizability of the findings may be limited to the study population (Koreans). For more definitive evidence on the effects of COVID-19 infection on the risk of uveitis, it is necessary to compare our results with those of individuals without COVID-19 infection, ideally matched with our population in terms of sex, age, and major confounding factors affecting uveitis recurrence, which could not be performed in this study. Assuming that uveitis recurrence is not frequent in most cases, such a comparison should be conducted in future studies with a large number of uninfected individuals. Also, future research should explore the immunological mechanisms underlying the associations among COVID-19 infection, vaccination, and uveitis recurrence, as well as the interaction between COVID-19 infection and vaccination, in different populations.

## 5. Conclusions

In conclusion, this study demonstrates a significantly increased risk of uveitis recurrence following COVID-19 infection, particularly in the early post-infection period. Although vaccination appears to attenuate the risk of uveitis recurrence, clinicians should be vigilant in monitoring for uveitis recurrence in recently infected individuals with a history of uveitis. Further research is needed to elucidate the underlying mechanisms and to optimize preventive measures to mitigate this ocular complication of COVID-19.

## Figures and Tables

**Figure 1 vaccines-12-00631-f001:**
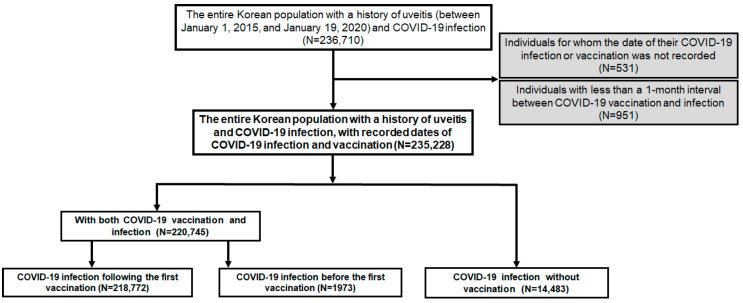
Flowchart of the study population and inclusion/exclusion criteria with our subject categorization scheme applied.

**Figure 2 vaccines-12-00631-f002:**
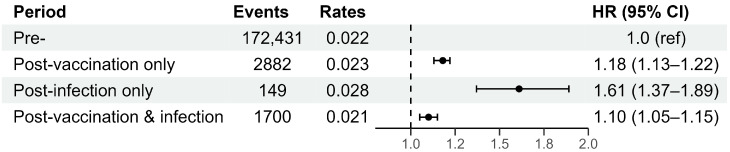
Risk of post-COVID-19 uveitis in various periods before and after COVID-19 vaccination and infection.

**Table 1 vaccines-12-00631-t001:** Demographic characteristics, comorbidities, and details of COVID-19 infection and vaccination among individuals with a history of uveitis (n = 235,228).

Characteristics	Value
**Age, mean (SD)**	55.6 (19.6)
<20	11,395 (4.8%)
20–39	40,070 (17.0%)
40–59	70,346 (29.0%)
60–79	89,273 (38.0%)
≥80	24,144 (10.3%)
**Sex**	
Male:female	108,393 (46.1%):126,835 (53.9%)
**Comorbidities**	
Hypertension	90,662 (38.5%)
Diabetes	67,977 (28.9%)
Rheumatic diseases	162,076 (68.9%)
**COVID-19 infection**	
**Number of infection event**	
One	230,793 (98.2%)
Two	4236 (1.8%)
Three or more	19 (<0.01%)
**Timing of infection**	
Before the first vaccination	16,456 (7.0%)
After the first vaccination	1993 (0.9%)
After the second (or subsequent) vaccination	216,779 (92.2%)
**COVID-19 vaccination**	
**Dose**	220,745 (93.8%)
First dose administered	220,745 (93.8%)
Second dose administered	218,972 (93.1%)
Third dose administered	177,047 (75.3%)
Fourth or further doses administered	74,878 (31.8%)
**Vaccine used for the first vaccination**	
**mRNA vaccine**	
BNT162b2 (Pfizer)	114,672 (52.0%)
mRNA-1273 (Moderna)	22,003 (10.0%)
**Adenovirus vector-based vaccine**	
ChAdOx1 (AstraZeneca)	78,082 (35.4%)
Ad26.COV2.S (Janssen)	5589 (2.5%)
**Intervals (days), m** **ean (median)**	
Between first and second vaccination	51 (42.0)
Between second and third vaccination	135.8 (122.0)
Between the latest vaccination and the first COVID-19 infection *	131.7 (113.0)

* Among the vaccinated population.

**Table 2 vaccines-12-00631-t002:** Hazard ratios (HRs) for pre- and post-COVID-19 uveitis. CI = confidence interval.

Period	Events	Rates *	HR (95% CI)
**Overall**			
**Pre-infection**	175,325	0.022	1.00 (ref)
**Post-infection**	1839	0.022	1.21 (1.07–1.37)
Early onset ^†^	231	0.025	1.42 (1.24–1.61)
Delayed onset ^†^	1608	0.021	1.09 (1.03–1.14)

* Person-months; ^†^ early-onset uveitis is defined as occurring within 30 days of the most recent infection, while delayed-onset uveitis is defined as occurring after this period.

**Table 3 vaccines-12-00631-t003:** Hazard ratios (HR) for pre- and post-COVID-19 uveitis in patients who received COVID-19 vaccinations or not at the time of infection. CI = confidence interval.

Period	Events	Rates *	HR (95% CI)
**COVID-19 infection after COVID-19 vaccinations**			
Pre-infection (after COVID-19 vaccination)	2882	0.023	1.00 (ref)
Post-infection (until the next COVID-19 vaccination/infection)	214	0.023	1.21 (1.05–1.39)
Early onset ^†^	213	0.025	1.21 (1.05–1.39)
Delayed onset ^†^	1	0.024	1.22 (0.17–8.64)
**COVID-19 infection before COVID-19 vaccinations**			
Pre-infection	1345	0.026	1.00 (ref)
Post-infection (until the next vaccination)	28	0.049	2.07 (1.42–3.02)
Early onset ^†^	4	0.072	3.61 (1.35–9.66)
Delayed onset ^†^	24	0.047	1.94 (1.29–2.90)

* Person-months; ^†^ early-onset uveitis is defined as occurring within 30 days of the most recent infection, while delayed-onset uveitis is defined as occurring after this period.

**Table 4 vaccines-12-00631-t004:** Cumulative incidences of post-COVID-19 uveitis at different time points.

Category	3 Months	6 Months	1 Year
**Post-COVID-19 uveitis**	19,946 (8.5%)	27,729 (11.8%)	32,795 (14.0%)
Anterior type	16,456 (7.0%)	22,943 (9.8%)	27,275 (11.6%)
Non-anterior type	3490 (1.5%)	4786 (2.0%)	5520 (2.4%)

**Table 5 vaccines-12-00631-t005:** Association of anatomic (anterior and non-anterior) types and etiologic (infectious and non-infectious) conditions between pre- and post-COVID-19 uveitis.

Pre-Infection Uveitis	Post-Infection Uveitis		*p*
*Anterior* vs. *non-anterior*	Anterior	Non-anterior	<0.001
Anterior	24,080	263	
Non-anterior	1837	4860	
*Infectious* vs. *non-infectious*	Infectious	Non-infectious	<0.001
Infectious	71	151	
Non-infectious	28	30,790	

## Data Availability

Data are unavailable due to privacy and ethical restrictions.

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
