# Peer review of "Risk of Post-COVID-19 Uveitis and Risk Modification by Vaccination: A Nationwide Retrospective Cohort Study"

_vaccines, 2024, doi:10.3390/vaccines12060631_

Round 1

Reviewer 1 Report

Comments and Suggestions for Authors

The introduction is appropriate and clearly states the background on which basis the study was conducted.

Methodology

1) It is not clear the period of time whose data have been provided by the KDCA. 

2) The first sentence should describe the study design, the place where the study was conducted and the time period. Please, add a sentence at the beginning of this section.

3) Ethical Committee approval should be at the end of the Methodology section. 

4) L79-95. The number of individuals identified is already a result: therefore it belongs to Results section. Only inclusion and exclusion criteria should be stated, not numbers.

5) L104. Please, change 'infection period' with 'pre-infection period', in order to make the concept clearer.

6) Considering definitions, a figure or a small box would give a more immediate insight. Not mandatory, just a suggestion.

7) L117-119. Were these data available from KDCA or authors got them through other databases/methodologies?

Results

All percentages are normally accompanied by crude numbers. As numbers are particularly high in this study, I would advise writing a sentence declaring crude numbers are displayed in Table 1.

In the first line of Results, show the selection process of data (filtered for exclusion and inclusion criteria).

L139-141. Should be moved after the sentence ending at L153. The same speech for Table 2- and Table 3-mentioning in the text. If authors prefer to keep the order they chose, it is fine anyway.

L141. It would be more fluent expressing the standard deviation in brackets.

Considering Table 1:
Please, try to divide visually COVID-19 infections from COVID-19 vaccines. Typo: Astrazeneca is misspelled.

Discussion is well argued and I only have a few comments. As far as I understood, the study is on the risk of recurrence of uveitis following COVID-19 infection or COVID-19 vaccination. In this section it is not clear that your cohort is composed by patients who already experienced a uveitis before COVID-19 infection/vaccination. Try to make it clearer to the reader.

L241. It is not clear what authors mean with the sentence 'This also indicates that the effect of vaccination, especially mRNA vaccines, on uveitis during the study period, amidst the COVID-19 pandemic, should be carefully discerned from that of COVID-19 infection itself'. Vaccination acceptance rate by uveitis-experienced patients does have little to do with the effect of vaccination itself, such as to advise a discrimination between the infection and the vaccine. Please, rephrase in order to get concepts tidier and more logic.

L249 'recurrence' instead of 'occurence'.

L259-289. The concept of COVID-19 vaccination is protective towards uveitis recurrence is clear but a little redundant. Try to reduce these lines to 15-20 lines maximum.

Conclusion

The association, for what I read, is between risk of uveitis recurrence and COVID-19 infection. Not the risk of a new uveitis acquisition. Please, check it and fix it. 

It would have been interesting to compare your results with uveitis recurrence risk associated to other viral infections in the Discussion section.

Author Response

Reviewer #1

The introduction is appropriate and clearly states the background on which basis the study was conducted.

→ Thank you for your thoughtful review and helpful suggestions. As per your comments, we have revised our manuscript by making appropriate corrections and clarifications.

Methodology

1) It is not clear the period of time whose data have been provided by the KDCA. 

→ Thank you for your comments. We clarified the data collection period in the manuscript. The data provided by the KDCA include the dates of COVID-19 infection and vaccinations from October 8, 2020, to December 31, 2022 (the end of our study). However, vaccination in Korea started on February 26, 2021, when the vaccination data were made available.

In the text, we added the sentence “the Korea Disease Control and Prevention Agency (KDCA) database, which provides details on COVID-19 infection and vaccinations, including types and timing of COVID-19 vaccine doses administered as well as the diagnosis and timing of COVID-19 infection between October 8, 2020, to December 31, 2022 (the end of our study)..” (page 2, line 69–73)

2) The first sentence should describe the study design, the place where the study was conducted and the time period. Please, add a sentence at the beginning of this section.

→ Thank you for your comments. We added the following sentence at the beginning of the section: "This is a retrospective study on a nationwide cohort with a history of uveitis and COVID-19 infection observed for the study period between January 1, 2015 and December 31, 2022." (page 2, lines 64-66)

3) Ethical Committee approval should be at the end of the Methodology section. 

→ We moved the sentences on ethical committee approval to the end of the Methods section. (page 4, lines 136-140)

4) L79-95. The number of individuals identified is already a result: therefore it belongs to Results section. Only inclusion and exclusion criteria should be stated, not numbers.

→ Thank you for your comments. We removed the number of individuals from the Methods section to ensure that only the inclusion and exclusion criteria are stated in the Methodology section. Instead, all numbers are stated in the Results section.

The lines in the Methods section now read:

Among the individuals with a history of uveitis and COVID-19 infection, those in whom the date of their COVID-19 infection or vaccination was not recorded were excluded. Additionally, individuals infected with COVID-19 within 1 month of vaccination, either before or after, were excluded to eliminate the combined acute effects of recent vaccination and infection exerted on the individuals, in whom discriminating each from the other would have proved impossible. After the exclusion, the study population consisted of the entire Korean population with a prior history of both uveitis and COVID-19 infection for whom the dates of infection and vaccination, separated by at least a 1-month interval, were both recorded.

For comprehensive analyses on the effect of COVID-19 infection, its vaccination, and both, the individuals were categorized as follows. First, those with COVID-19 infection were separated into those with both COVID-19 vaccination and infection and those with infection only. Second, the individuals with both vaccination and infection were divided into two groups, infection before 1st vaccination and infection following 1st vaccination. Figure 1 provides a graphical summary of the inclusion/exclusion criteria and our subject categorization scheme.

→ In the Results section, we added the sentences “Among the 236,710 individuals with a history of uveitis and COVID-19 infection, 235,228 were included in our analyses after applying the exclusion criteria (Figure 1). The cohort comprised 220,745 individuals who had both COVID-19 vaccination and infection and 14,483 who had the infection only.” (page 4, lines 143–146)

5) L104. Please, change 'infection period' with 'pre-infection period', in order to make the concept clearer.

→ Thank you for your suggestion. We changed the phrase ‘infection period’ to ‘pre-infection period’ per your suggestion. (line 104 of the revised manuscript).

6) Considering definitions, a figure or a small box would give a more immediate insight. Not mandatory, just a suggestion.

→ Supplementary Table S1 was used for definitions (uveitis types). As per your suggestion, we have added the other definitions in the table to provide more immediate and clearer insight to the readers.

7) L117-119. Were these data available from KDCA or authors got them through other databases/methodologies?

→ All data publicly available from KDCA and NHIS, were provided by both government agencies upon request. In the sentence, we have indicated that these data were provided by the KDCA and NHIS.

In the text, we have added the sentences “We utilized the combined data from the two databases: the Korean National Health Insurance Service (NHIS) database, which includes the entire Korean population diagnosed with uveitis between January 1, 2015 and January 19, 2020 (the day before the first COVID-19 infection in Korea), and the Korea Disease Control and Prevention Agency (KDCA) database, which provides details on COVID-19 infection and vaccinations, including types and timing of COVID-19 vaccine doses administered as well as the diagnosis and timing of COVID-19 infection between October 8, 2020, to December 31, 2022 (the end of our study). Other information on demographic data, inpatient and outpatient health care visits, prescription medications, and diagnoses were provided by the NHIS.” (lines 66-75)

Results

All percentages are normally accompanied by crude numbers. As numbers are particularly high in this study, I would advise writing a sentence declaring crude numbers are displayed in Table 1.

→ Thank you for your helpful suggestion. We added a sentence to indicate that crude numbers are displayed in Table 1 as follows: Both the crude number of individuals and their corresponding percentages are included. (lines 149–150)

In the first line of Results, show the selection process of data (filtered for exclusion and inclusion criteria).

→ Yes. We added the first paragraph describing the number of individuals filtered for inclusion and exclusion criteria as follows: Among the 236,710 individuals with a history of uveitis and COVID-19 infection, 235,228 were included in our analyses after applying the exclusion criteria (Figure 1). The cohort comprised 220,745 individuals who had both COVID-19 vaccination and infection and 14,483 who had the infection only. (lines 144-147)

L139-141. Should be moved after the sentence ending at L153. The same speech for Table 2- and Table 3-mentioning in the text. If authors prefer to keep the order they chose, it is fine anyway.

→ We appreciate your feedback regarding the organization of the text. However, we believe that maintaining the current structure with an introductory summary sentence (referencing the table number) helps readers grasp the content more effectively. We will continue to use this style unless it conflicts with the journal's guidelines. Thank you for the suggestion.

L141. It would be more fluent expressing the standard deviation in brackets.

→ Per your suggestion, we changed the sentence to “The mean (SD) age of the participants was 55.6 (19.6) years, highlighting a diverse age distribution.” (lines 150–151)

Considering Table 1:
Please, try to divide visually COVID-19 infections from COVID-19 vaccines. Typo: Astrazeneca is misspelled.

→ We apologize for this typographical error. The error has been rectified and to enhance clarity, we have visually separated COVID-19 infections from vaccination details by (1) adding a horizontal line between them and (2) introducing additional subheadings titled 'COVID-19 Infection' and 'COVID-19 Vaccination'.

Discussion is well argued and I only have a few comments. As far as I understood, the study is on the risk of recurrence of uveitis following COVID-19 infection or COVID-19 vaccination. In this section it is not clear that your cohort is composed by patients who already experienced a uveitis before COVID-19 infection/vaccination. Try to make it clearer to the reader.

→ We updated the term "uveitis" to "uveitis recurrence" where appropriate throughout the Discussion section to highlight that our study focused on the risk of uveitis recurrence following COVID-19 infection or COVID-19 vaccination.

L241. It is not clear what authors mean with the sentence 'This also indicates that the effect of vaccination, especially mRNA vaccines, on uveitis during the study period, amidst the COVID-19 pandemic, should be carefully discerned from that of COVID-19 infection itself'. Vaccination acceptance rate by uveitis-experienced patients does have little to do with the effect of vaccination itself, such as to advise a discrimination between the infection and the vaccine. Please, rephrase in order to get concepts tidier and more logic.

→ Thank you for your insightful comments. We revised this sentence to enhance clarity and coherence. The updated sentence now reads as follows: "This also suggests that distinguishing between the impact of vaccination and COVID-19 infection on uveitis recurrence among the included patients with COVID-19 infection is crucial, considering that a vast majority of these patients received vaccination during the study period." (lines 261–264)

L249 'recurrence' instead of 'occurence'.

→ We changed the word ‘occurrence’ to ‘recurrence’ as per your suggestion.

L259-289. The concept of COVID-19 vaccination is protective towards uveitis recurrence is clear but a little redundant. Try to reduce these lines to 15-20 lines maximum.

→ By removing redundant sentences, we reduced the text to 20 lines as follows: Furthermore, our findings for the different post-infection and vaccination periods elucidate the differential impact of vaccination and infection on the risk of post-COVID-19 uveitis. The significantly elevated HRs during the post-infection-only period (1.61) highlight the acute, severe intraocular inflammation, as is typically observed with systemic inflammation following COVID-19 infection. Conversely, the slightly elevated HRs observed during the post-vaccination-only period (1.18) suggest an association between vaccination and uveitis, albeit to a lesser extent compared with infection. This is consistent with previous findings showing significant associations between COVID-19 vaccination and uveitis recurrence for people with a history of uveitis. [10-12,16] The slightly elevated HR during the post-vaccination and infection period (1.10), similar to that during the post-vaccination-only period, may indicate a much-mitigated host response after COVID-19 infection in vaccinated individuals, leading to reduced risk of uveitis recurrence.

While concerns have been raised regarding uveitis recurrence following vaccination, our results suggest that vaccination may actually mitigate the risk of uveitis recurrence in individuals with a history of uveitis who contract COVID-19. These findings underscore the importance of balanced risk assessment and informed decision-making regarding vaccination, particularly for more vulnerable populations such as those with recurrent uveitis. Therefore, our study contributes to the ongoing dialogue surrounding vaccination strategies for uveitis patients and potentially suggests the role of vaccination in alleviating the overall burden of COVID-19-related complications, including post-infection uveitis. (lines 280–299)

Conclusion

The association, for what I read, is between risk of uveitis recurrence and COVID-19 infection. Not the risk of a new uveitis acquisition. Please, check it and fix it. 

→ Thank you for your careful review and feedback regarding our manuscript. We reviewed the conclusion and made the necessary adjustments. The Conclusion section now reads:

  1. Conclusions

In conclusion, this study demonstrates a significantly increased risk of uveitis recurrence following COVID-19 infection, particularly in the early post-infection period. Although vaccination appears to attenuate the risk of uveitis recurrence, clinicians should be vigilant in monitoring for uveitis recurrence in recently infected individuals with a history of uveitis. Further research is needed to elucidate the underlying mechanisms and optimize preventive measures to mitigate this ocular complication of COVID-19. (lines 358-364)

It would have been interesting to compare your results with uveitis recurrence risk associated to other viral infections in the Discussion section.

→ We appreciate your suggestion and incorporated the following paragraph into the Discussion section:

Moreover, comparing the risk of uveitis recurrence associated with COVID-19 with that associated with other systemic viral infections could offer valuable insights into the distinct characteristics of post-COVID-19 uveitis. Although recurrent uveitis following other viral infections has been documented, existing reports predominantly focus on infectious uveitis resulting from viral infections, particularly those of the Herpesviridae family, such as cytomegalovirus, herpes simplex virus, and varicella-zoster virus.[18-21] One population-based study showed a significant association between herpes zoster and the subsequent risk of anterior uveitis, with a hazard ratio (HR) of 1.67 during the 1-year follow-up period following zoster infection, increasing to 13.06 for herpes zoster ophthalmicus.[19] However, systematic evaluations of post-infection uveitis in the context of other viral infections have been limited because of the smaller patient populations studied, unlike for COVID-19, which has affected a large number of individuals globally as a pandemic. Future studies should investigate how the risk of uveitis recurrence is associated with COVID-19 compared with that of other viral infections, including herpes viruses or other viruses known to induce ocular inflammation.[18-21] Understanding the relative risk posed by COVID-19 compared to other viral infections could illuminate potential shared mechanisms or distinct pathophysiological pathways underlying post-viral uveitis. Furthermore, comparative analyses may inform clinical management strategies and guide tailored preventive measures specific to different viral etiologies. (lines 313-331)

Reviewer 2 Report

Comments and Suggestions for Authors

After reviewing this manuscript, how many people who didn’t get covid but had a history of uveitis had a uveitis recurrence during the time period, especially since this reports shows that both being vaccinated and getting 1 or more COVID infections increase the risk of uveitis (figure 2).  I understand that you excluded these people from your analysis.  Presumably you have those data as well?  As it now stands, your interpretations are based on the assumption that they don’t naturally recur. Add to the discussion if you can’t extend the analyses?

Did you count only first vaccination date? – methods say yes and most got covid after 2nd vaccination – do we know time interval from last vaccination? (see table 1).  It seems that a time to event analysis would be more appropriate to characterize the events after vaccination or booster; otherwise we are left wondering if the first vaccination might have been months or years before the uveitis recurrence.  Table 3 should be updated to examine covid-19 infection after latest vaccination, or define a consistent observation period for follow-up. 

Your time periods for classifying early and late occurrence are not consistent.  I don’t see a definition of what constitutes early onset uveitis or delayed onset, as shown in tables 2 and 3, nor how this relates to the timing definitions used in table 4 (<3 months, at 6 months and at 1 yr post infection.  At the very least, please  add a footnote or otherwise mention your definition of “early onset” and “delayed onset”

 Discussion – 2nd paragraph refers to conclusions relating to mRNA vaccines.  It would be helpful to mention whether all or if not, which vaccines, listed in Table 1 are mRNA vaccines

Author Response

Reviewer #2

After reviewing this manuscript, how many people who didn’t get covid but had a history of uveitis had a uveitis recurrence during the time period, especially since this reports shows that both being vaccinated and getting 1 or more COVID infections increase the risk of uveitis (figure 2).  I understand that you excluded these people from your analysis.  Presumably you have those data as well?  As it now stands, your interpretations are based on the assumption that they don’t naturally recur. Add to the discussion if you can’t extend the analyses?

→ Thank you for your insightful review and helpful suggestions. For this study, we received data for individuals with COVID-19 infection from the KDCA and NHIS, as those without the infection were not the focus of our analysis. Instead of using individuals without infection as a control group, we compared pre- and post-COVID-19 infection within the same individuals. Different patients (those without COVID-19) may have different demographic and clinical characteristics (e.g., age, sex, and underlying diseases), which may also affect uveitis recurrence. In contrast, comparing before and after COVID-19 infection within the same individuals can control for potential confounding factors, such as demographic and clinical characteristics, resulting in more reliable results regarding the effect of COVID-19 infection on uveitis.

Nevertheless, we strongly agree with your point on the need for data from patients without COVID-19 infections, but we could not obtain such data as it was not included in what we received for this study. However, as you pointed out, assuming that uveitis recurrence is uncommon, it is necessary to compare our results to those without COVID-19 infection, ideally matched with our population in terms of sex, age, and major confounding factors affecting uveitis recurrence. Accordingly, we have briefly addressed the relevant discussion to the Discussion section that this should be addressed in future studies as follows:

For more definitive evidence on the effect of COVID-19 infection on the risk of uveitis, it is necessary to compare our results with those of individuals without COVID-19 infection, ideally matched with our population in terms of sex, age, and major confounding factors affecting uveitis recurrence, which could not be done in this study. Assuming that uveitis recurrence is not frequent in most cases, such a comparison should be conducted in future studies with a large number of uninfected individuals. (Page 11, Lines 348-353)

Did you count only first vaccination date? – methods say yes and most got covid after 2nd vaccination – do we know time interval from last vaccination? (see table 1).  It seems that a time to event analysis would be more appropriate to characterize the events after vaccination or booster; otherwise we are left wondering if the first vaccination might have been months or years before the uveitis recurrence. Table 3 should be updated to examine covid-19 infection after latest vaccination, or define a consistent observation period for follow-up.

→ Thank you for your insightful comments. Previous studies have shown that the effect of vaccination on uveitis is the greatest after the first vaccination. (Kim J. et al. 2024, Tomkins-Netzer et al. 2022, and Jordan et al. 2023) Although its acute (i.e., 30 days within vaccination) effect is greater than the delayed (i.e., after 30 days) effect, delayed onset uveitis following the first vaccination cannot be neglected. Thus, even in cases of uveitis recurrence occurring after the 2nd vaccination, the effect of the 1st vaccination on uveitis cannot be neglected. Therefore, we performed a time-to-event (survival) analysis by defining post-vaccination uveitis as that after the first vaccination and calculated HR for post-vaccination uveitis using Cox proportional hazards model. Although this has limitation that post-vaccination period is heterogeneous in terms of uveitis risk after subsequent vaccinations, despite smaller effect than the first vaccination in previous studies, it is simple and straightforward and represents the effect of the most significant event in terms of vaccination.

In response to your comment, we thought that we need to additionally perform a time-to-event analysis to provide descriptive data on the intervals (1) between first and second vaccinations, (2) between second and third vaccinations, and (3) between latest vaccination and COVID-19 infection in vaccinated population, to better characterize the time to event following vaccination or booster doses. The data added to Table 1 also provide a more accurate depiction of the timing between vaccination and uveitis recurrence and interval between vaccinations. Additionally, we have updated Table 3 to examine post-COVID-19 uveitis occurring after COVID-19 infection until the next vaccination, ensuring a consistent observation period for follow-up.

In the Discussion section, we have added the limitation “The post-vaccination period might be heterogeneous in terms of uveitis risk owing to subsequent vaccinations despite having a smaller effect than the first one in previous studies.[12,13,16] However, the post-vaccination period in this study, rather than the post-first vaccination or post-second vaccination period, is simple and straightforward and represents the effect of the most significant event in terms of vaccine-associated uveitis, the first vaccination.” (Page 10, Lines 336-341)

Your time periods for classifying early and late occurrence are not consistent.  I don’t see a definition of what constitutes early onset uveitis or delayed onset, as shown in tables 2 and 3, nor how this relates to the timing definitions used in table 4 (<3 months, at 6 months and at 1 yr post infection.  At the very least, please add a footnote or otherwise mention your definition of “early onset” and “delayed onset”

→ The distinction between early- and delayed-onset uveitis has been used in previous studies. (Kim J, et al. and Chang MS, et al.) These are typically used for uveitis following COVID-19 infection or vaccination.

Cumulative incidences were investigated for 3 and 6 months and 1 year, which are commonly used in the ophthalmology literature. In Tables 2 and 3, we added footnotes “Early-onset uveitis is defined as occurring within 30 days of the most recent infection, while delayed-onset uveitis is defined as occurring after this period.” to define early and delayed onset.

 Discussion – 2nd refers to conclusions relating to mRNA vaccines.  It would be helpful to mention whether all or if not, which vaccines, listed in Table 1 are mRNA vaccines

→ We strongly agree with your point. BNT162b2 and mRNA-1273 are mRNA vaccines. However, as per Reviewer #1’s comments, we revised the sentence, and the word ‘mRNA vaccines’ is no longer included in the Discussion section. However, additional information on the vaccine types has been added to Table 1.

Reviewer 3 Report

Comments and Suggestions for Authors

The article entitled Risk of Post-COVID-19 Uveitis and Risk Modification by Vaccination: A Nationwide Retrospective Cohort Study is very clearly written. A selected national cohort of patients with uveitis in the pre-pandemic period, a set of variables studied in relation to infection with SARS-CoV-2 and vaccination against COVID-19 was described. The conclusions of the research are very important for understanding the effect of SARS-CoV-2 on eye complications, as well as regarding the adverse effects of vaccination in people who have recovered from uveitis.

I suggest minor corrections or improvements:

1. There is no specified  period of data collection regarding SARS-CoV-2 infection or vaccination  in persons with uveitis - was it until the end of 2023?

2. You collected data on age and gender of patients with uveitis - were there differences in the incidence of post-infectious uveitis or post-vaccination uveitis by age group or by gender?

Author Response

Reviewer #3

The article entitled Risk of Post-COVID-19 Uveitis and Risk Modification by Vaccination: A Nationwide Retrospective Cohort Study is very clearly written. A selected national cohort of patients with uveitis in the pre-pandemic period, a set of variables studied in relation to infection with SARS-CoV-2 and vaccination against COVID-19 was described. The conclusions of the research are very important for understanding the effect of SARS-CoV-2 on eye complications, as well as regarding the adverse effects of vaccination in people who have recovered from uveitis.

I suggest minor corrections or improvements:

  1. There is no specified period of data collection regarding SARS-CoV-2 infection or vaccination in persons with uveitis - was it until the end of 2023?

Thank you for your review and helpful suggestions. We added information on the period of data collection regarding COVID-19 infection and vaccination as follows: the Korea Disease Control and Prevention Agency (KDCA) database, which provides details on COVID-19 infection and vaccinations, including types and timing of COVID-19 vaccine doses administered as well as the diagnosis and timing of COVID-19 infection between October 8, 2020, to December 31, 2022 (the end of our study)” (lines 69–73)

  1. You collected data on age and gender of patients with uveitis - were there differences in the incidence of post-infectious uveitis or post-vaccination uveitis by age group or by gender?

Thank you for your insightful comments. We analyzed the data by separating the patients into four age and sex groups. Our findings indicate a slightly increased risk of post-COVID-19 uveitis in older age groups. However, no significant differences were observed between male and female patients in terms of the risk of post-infection uveitis. These results were incorporated into supplementary tables to provide a more detailed understanding of the demographic influences on uveitis recurrence following COVID-19 infection.

In the text, we added the following paragraph: “In the subgroup analyses according to age and sex, a similar trend of the highest risk of uveitis in the post-infection only period was noted across all age and sex groups. However, a slight increase in the risk of post-COVID-19 uveitis was observed in older age groups (Supplementary Table S2), whereas no remarkable difference in post-infection uveitis was observed between male and female patients (Table S3).” (lines 218-222)

Round 2

Reviewer 2 Report

Comments and Suggestions for Authors

for Table 1, please add the notation that the row titled Intervals shows DAYS.

Otherwise, good revision.  thanks for your attentiveness to my suggestions.

Author Response

Author’s Response to Reviewer’s Comment

Reviewer #2

for Table 1, please add the notation that the row titled Intervals shows DAYS.

Otherwise, good revision.  thanks for your attentiveness to my suggestions.

→ Thank you for your feedback. We've made the necessary adjustments to Table 1, incorporating the notation that the row labeled Intervals represents days. Your meticulous attention to detail is greatly appreciated, and we're pleased to hear that you're satisfied with the revisions. The heading now accurately reads as follows: Intervals (days), mean (median).
